# Optimization of Cargo Shipping Adaptability Modeling Evaluation Based on Bayesian Network Algorithm

**Siyuan Gao** [1,2], **Fengrong Zhang** [3,*], **Wei Ning** [4] **and Dayong Wu** [5]

1 School of Economics and Management, Northeast Normal University, Changchun 130024, China
2 Faculty of Education, Northeast Normal University, Changchun 130024, China
3 Faculty of Marxism, Northeast Normal University, Changchun 130024, China
4 Economic Research Institute of Jilin Province Development and Reform Commission, Changchun 130051, China
5 Texas A&M Transportation Institute, Texas A&M University, College Station, TX 77843, USA
* Correspondence: zhangfr631@nenu.edu.cn

**Abstract:** Through shipping service adaptability measurement, selecting shipping services that are more adaptable to preferences such as low cost, high efficiency, safety, and obvious emission reduction can achieve synergistic optimization of green shipping management. The study takes green shipping service adaptability as the research theme; explores three aspects, i.e., shipping safety, shipping rate and shipping choice preference, related to the evaluation and selection of a green shipping service; constructs the green shipping service adaptability evaluation index system including safety index, freight rate index and choice preference index; and applies fuzzy-exact by processing the historical data from H shipping company in Hainan Province, China. Bayesian net is applied to calculate the shipping safety adaptation degree of the transportation object. The theory of shipping service adaptability proposed in the paper can be applied to the fields of shipping supplier selection and shipping company's detection of shipping object status. The fuzzy-exact Bayesian network method chosen in the paper can solve the problem of incomplete state coverage of the Bayesian network and correct the situation that some edge probabilities are unreasonable.

**Keywords:** cargo shipping services; green shipping; adaptability; Bayesian networks; evaluation system design

## 1. Introduction

Water transportation is one of the five main modes of transportation and has advantages of large capacity, low price and low energy consumption, so it occupies an important position in the transportation of passengers and goods. However, it has many shortcomings, such as slow transportation speed, seasonality, port availability, climate, interrupted time, etc. These shortcomings have limited its development. With global energy shortages and increasing environmental concerns in various countries [1,2], green environmental protection has become the main tone of shipping industry development. Green shipping considers the perspective of the environment and sustainable development, not only during transport but also emphasizing shipping safety, transport price and the environment, in harmony with each other. The most general understanding of "green shipping" assumes the use of resources and energy by different types of both cargo and passenger vessels in such a way that prevents pollution and reduces shipping impact on the global environment [3–6]. Green shipping has a huge impact on the structure of sustainable economic and environmental performance [7,8]. It has also been shown that the development of green shipping enhances the competitiveness of ports and other participants in shipping infrastructure [9]. Therefore, it can better achieve sustainable development. Under the influence of the current COVID-19 pandemic and with the rapid development of the economy and the shortening of the product life cycle [10], the necessity of the concept of "green shipping" has been highlighted even more [2,11]. The application of modern shipping

science and technology to shipping and daily management has the aim of achieving green and efficient cargo shipping services. Research in shipping has focused mainly on transport safety and satisfaction, while little research has been carried out on the adaptability of shipping services. When choosing a shipping service, the transport object not only has the requirement of accessibility, but also has other demands, such as the price of shipping, safety, service level, green shipping, etc. Shipping service adaptability is an important index to examine the adaptability of shipping services to the transportation object. Shipping service adaptability takes into account the shipping safety of cargo, shipping price and customer preference, and can help cargo owners select shipping suppliers. In this paper, our research into cargo shipping service adaptability includes three aspects: cargo shipping safety, price and choice preference. Through research into these three aspects, an environmental symbiotic shipping management system was established from the perspective of environmental and sustainable development and a shipping mode representing sustainable development and environmental protection was realized. In this way, cargo shipping is successfully adapted to the development of green shipping [12,13].

### 1.1. Adaptability and Adaptability Studies

Adaptation is a concept in biology which is typically applied in the research field of animals and microorganisms. The concept of adaptation in biology involves the survival rate and reproduction ability of alleles, individuals, or groups in evolution. Many studies have been carried out on the adaptation of organisms abroad and the main research direction has been focused on the adaptation of animals to the environment [14,15]. There are also studies on the adaptation of microorganisms to the environment [16]. In contrast, the research on adaptation in the field of transportation includes the following main areas.

(1)    Adaptation of transport construction to economic development:

Brown provided insight into the relevance of transport infrastructure and its assessment through his study of complex adaptive systems, arguing that individuals, manufacturing systems and nations are dependent on transport infrastructure, and that this is adaptive [17]. Sussman suggested that transport development planning should be seen as a system that needs to consider the complex factors involved and the adaptability of transport construction to social, economic, and environmental development, and that it should be discussed in depth from two entry points: institutional integration and technological progress [18]. Knoflacher analyzed the historical evolution of transport planning in Europe and summarized the relevant experiences, suggesting that transport planning is not only a transport issue, but is also related to the urban environment and socioeconomics [19].

(2)    Adaptations in traffic accident analysis and prevention:

In conjunction with driving adaptability, Langford illustrated the relationship between driving miles and accident rates for drivers with many years of driving experience, showing that the shorter the driving miles, the higher the accident rate, corresponding to poorer driving adaptability and susceptibility to accidents [20]. Marino studied the influence of chronic diseases on drivers' driving adaptability [21].

In addition to this, there is a small body of adaptive research on supply chains. Brintrup studied the adaptive behavior of multi-agent supply chains and extended it to multi-objective, multi-functional supply chains on this basis [22]. Ivanov studied the adaptability of multi-agent supply chains through adaptive research on supply chain planning and scheduling using optimal control theory [23].

### 1.2. Bayesian Networks

The British mathematician Bayes in the 1860s delivered a paper on solving odds problems, which was the source of Bayesian nets. In the 1980s, Pearl first proposed Bayesian methods, while introducing them to expert systems [24,25]. In recent years, the study of Bayesian networks has become a very large area of research, both in terms of theoretical expansion and algorithm design, and Bayesian networks have made great progress [26,27].

To reveal the fire damage of warships, Jia et al. used Bayesian networks for risk assessment of warship fires caused by non-contact explosions [28]. Goerlandt et al. developed a probabilistic model of cargo oil outflow from a product tanker during a ship collision based on Bayesian networks to assess the environmental impact of the event [29]. To establish effective safety measures in order to avoid accidents, Wang et al. developed a human factor analysis and classification system using Bayesian nets and analyzed the accident model [30]. Hanninen et al. used port inspection data to model Bayesian networks for ship accidents [31]. In addition, Hanninen et al. evaluated the influence of human factors on ship collisions, and analyzed data mainly from the Gulf of Finland [32]. Zhang et al. recently integrated safety assessment methods and Bayesian network methods to evaluate the shipping risks in the Yangtze River and analyzed the accident probability and consequences using a risk matrix [33]. Chen developed a quantitative fuzzy causal model to evaluate the human–machine–environment system in hazard analysis and compared it with a Bayesian net, and the results showed that the fuzzy model was more suitable for uncertain hazard assessment [34]. For small-sample Bayesian nets, Efron et al. proposed the self-help method (bootstrap), a classical method for solving small-sample datasets, in 1993 in their introductory book [35]. The proposal of the self-help method made an important contribution to later problems of dealing with small sample data sets in statistics. The literature proposed a solution for Bayesian network learning with small data sets in 1999, applying the self-help method approach to the Bayesian network learning process for small data sets [36].

According to the above analysis of scholars' research on Bayesian networks, it could be seen that the applications of Bayesian networks were all mainly focused on the analysis of accidents, including the analysis of shipping accidents, road traffic accidents and electric power accident systems, mainly applying two methods, objective Bayesian networks in the case of sufficient data and subjective Bayesian networks with the application of expert systems. In the case of insufficient data volume, the main application now is the method of expanding the sample after sampling by the self-help method to meet the demand of data volume. Therefore, this paper proposes the calculation process of fuzzy-accurate Bayesian net by analyzing the calculation process of Bayesian net and fuzzy-set theory and applies fuzzy-accurate Bayesian net to calculate the shipping safety adaptation degree of transportation objects.

*1.3. Network Analysis Method*

In 1996, Professor Satty proposed another new approach to decision making based on hierarchical analysis, namely network analysis [37–39]. This method can combine qualitative and quantitative analysis to deal with decision problems from a system perspective and is widely used in the study of decision problems in various fields because of its many advantages. Lee and Kim applied network analysis to study the problem of information system selection goal planning [40]. Karsak and Sozer applied network analysis to explore the optimization problem of how a product is configured for its functions [41]. Gencer and Gurpinar applied network analysis to the supplier selection problem and studied the supplier selection of a company as an example [42]. Meade and Presley applied network analysis to the evaluation of project proposals and gave a case study [43]. Chung and Lee studied the application of network analysis to the product production mix problem of a semiconductor manufacturer [44].

In 2003, Super Decision Software Super Decisions V2.10was introduced in the United States which programmed the calculation of ANP based on the ANP theory and solved the problem of calculating the ANP judgment matrix [45]. For example, Liu and Zhang compared the ANP method with the AHP method, concluded that the ANP method increased the feasibility and rationality of indicator weights and applied the ANP method to evaluate the sustainable development of enterprises [46]. Bi et al. constructed an evaluation index system of informatization development in Tianjin and applied the ANP method to calculate the index system in order to obtain the evaluation results for the

informatization level in Tianjin [47]. Qian et al. applied Porter's diamond theory model to construct an evaluation index system for the competitiveness of the cultural industry and evaluated this using network hierarchy analysis [48].

Analyzing the status of scholars' research on network analysis method, we can see that it is a research method that better combines subjective method and objective method, so that scholars complete more research on this aspect; it was mainly applied in decision making research and the recent research method is mainly combined with the fuzzy comprehensive evaluation method to form the fuzzy network analysis method for decision making research. Therefore, this paper applies the network analysis method to rank the affiliation degree of each factor affecting the adaptability of shipping service in the established index system, eliminates the indexes with smaller influence according to expert suggestion, and simplifies the index system of shipping service adaptability.

## 2. Materials and Methods

### 2.1. Design of the Evaluation Indicator System

In green shipping, whether the shipping service is adapted to the transport object has a significant relationship with whether the cargo owner is satisfied with the shipping service. Customer satisfaction theory evaluates customer satisfaction from three levels: product, service and society. By considering green development and the theory of sustainable development as the basis, the analysis of customer satisfaction theory and the characteristics of shipping relate cargo shipping service adaptability to three aspects; namely, safety, freight price and service (choice preference). Therefore, the cargo shipping evaluation index system established in this paper mainly focuses on these three factors and 55 factors were obtained through the analysis of the influencing factors of cargo shipping adaptability to establish the cargo shipping service adaptability evaluation index system, as shown in Table 1.

**Table 1.** Cargo shipping service adaptability rating indicator system.

| System Level | Guideline Level | Code Breakdowns | Factor Layer |
|---|---|---|---|
| Cargo Shipping Security Adaptation D | Crew $D_1$ | Personality trait $D_{11}$ | Accountability $d_1$ |
| | | | Psychological quality $d_2$ |
| | | | Security awareness $d_3$ |
| | | Personal competence $D_{12}$ | Academic qualifications $d_4$ |
| | | | Operational capability $d_5$ |
| | | | Length of service on board $d_6$ |
| | | | Learning ability $d_7$ |
| | | | Attend training $d_8$ |
| | | Physiological conditions $D_{13}$ | Age $d_9$ |
| | | | Health status $d_{10}$ |
| | | | Fatigue level $d_{11}$ |
| | | Motivation $D_{14}$ | Love for the job $d_{12}$ |
| | | | Satisfaction with treatment $d_{13}$ |
| | Ship $D_2$ | Ship Maintenance $D_{21}$ | Ship maintenance levels $d_{14}$ |
| | | Age of vessel $D_{22}$ | Ship age status $d_{15}$ |
| | | Vessel tonnage $D_{23}$ | Vessel tonnage status $d_{16}$ |
| | | Hull structure $D_{24}$ | Structural stability of ships $d_{17}$ |
| | | | Hull strength $d_{18}$ |
| | | | Cargo requirements for ship construction $d_{19}$ |

**Table 1.** *Cont.*

| System Level | Guideline Level | Code Breakdowns | Factor Layer |
|---|---|---|---|
| | | Marine equipment $D_{25}$ | Communications signaling equipment $d_{20}$ |
| | | | Lashing equipment $d_{21}$ |
| | | | Fire protection systems $d_{22}$ |
| | | | Cargo requirements for ship equipment $d_{23}$ |
| | Environment $D_3$ | Waterway conditions $D_{31}$ | Waterway markings $d_{24}$ |
| | | | Channel width $d_{25}$ |
| | | | Remaining water depth $d_{26}$ |
| | | Hydrological conditions $D_{32}$ | Water velocity $d_{27}$ |
| | | | Direction of water flow $d_{28}$ |
| | | | Wave High $d_{29}$ |
| | | Meteorological conditions $D_{33}$ | Wind $d_{30}$ |
| | | | Visibility $d_{31}$ |
| | | | Temperature $d_{32}$ |
| | | | Air humidity $d_{33}$ |
| | | Navigation order $D_{34}$ | Vessel Density $d_{34}$ |
| | | | Waterway Order $d_{35}$ |
| | Management $D_4$ | Security Management System $D_{41}$ | Soundness of safety management system $d_{36}$ |
| | | Safety Organization Training $D_{42}$ | Degree of safety organization training $d_{37}$ |
| | | Safety Promotion $D_{43}$ | Degree of security promotion $d_{38}$ |
| | | Emergency rescue system $D_{44}$ | Emergency rescue system sound and implementation $d_{39}$ |
| | | Enforcement of safety laws and regulations $D_{45}$ | Implementation of security laws and regulations level $d_{40}$ |
| | | Shipping Safety Information Technology $D_{46}$ | Shipping safety information level $d_{41}$ |
| | | Cargo stowage software application $D_{47}$ | The degree of application of cargo accumulation software $d_{42}$ |
| | | Cargo shipping regulations $D_{48}$ | Soundness of cargo shipping regulations $d_{43}$ |
| | Cargo characteristics $D_5$ | Cargo time $D_{51}$ | Cargo time security degree $d_{44}$ |
| | | Cargo loading location $D_{52}$ | Degree of security of loading position $d_{45}$ |
| | | Characteristics of the cargo itself $D_{53}$ | Physical and chemical characteristics of the cargo $d_{46}$ |
| | | Characteristics of cargo packaging $D_{54}$ | Degree of packaging to meet requirements $d_{47}$ |
| Cargo shipping tariff adaptability E | Freight rate $E_1$ | Vessel tariff $E_{11}$ | Vessel tariff $e_1$ |

**Table 1.** *Cont.*

| System Level | Guideline Level | Code Breakdowns | Factor Layer |
| --- | --- | --- | --- |
| Cargo Shipping Choice Preference Adaptation F | Convenience $F_1$ | Shipping interval $F_{11}$ | Shipping interval $f_1$ |
| | | Convenience of switching transportation modes$F_{12}$ | Ease of switching modes of transportation $f_2$ |
| | | Port Location Convenience $F_{13}$ | Port location convenience $f_3$ |
| | Efficiency $F_2$ | Transportation speed $F_{21}$ | Transportation speed $f_4$ |
| | Transport consistency $F_3$ | Transport consistency $F_{31}$ | Transport consistency$f_5$ |
| | Corporate image $F_4$ | Enterprise size $F_{32}$ | Enterprise size $f_6$ |
| | | Management standardization $F_{33}$ | Management standardization $f_7$ |

*2.2. Identification of Key Indicators*

Many nodes were obtained in this article which are difficult to calculate, and some have only a small impact on shipping service adaptability. This article applied the network analysis method to identify the indicators obtained in 2.1 and eliminate the factors that have only a small impact on shipping service adaptability.

2.2.1. Basic Principles of Network Analysis Method

In the mid-1980s, Saaty proposed the feedback AHP, which is the precursor of ANP and in 1996 Saaty proposed the theory and method of ANP more systematically on ISAHP-IV [37–39].

The decision-making principle of network analysis is basically the same as that of hierarchical analysis, the only difference being that the former builds a network structure model, while the latter builds a hierarchical structure model. Since the network structure model is far more complex than the hierarchical model, the network analysis method applies a more advanced mathematical knowledge in terms of weight synthesis, where the more important concept is the application and analysis of the supermatrix.

The network in the network analysis method consists of components and the influence between connected components; where the components in turn consist of the elements that make up the components; in the components, the elements can exist to interact with each other and with the elements in other components. In the network analysis method, the mutual influence relationship is expressed by the symbol "→", for example, "A→B" means that component B (element) is influenced by component A (element), or component A (element) influences component B (element), and the influence here mainly refers to the importance of the influence. In particular, the relationship of the influence of the component itself on itself is called the "feedback relationship" [49].

ANP first divided the system elements into two main parts, the first being the control factor layer, which included the problem objectives and decision criteria. All decision criteria can be considered independently of each other and were governed only by the goal element. The control layer must have an objective, but the decision criterion was not required. The weights of the criteria in the control layer were generally obtained by the AHP method. The second part, the network layer, was composed of the control layer's dominating elements and these elements interacting with each other. This is shown in Figure 1.

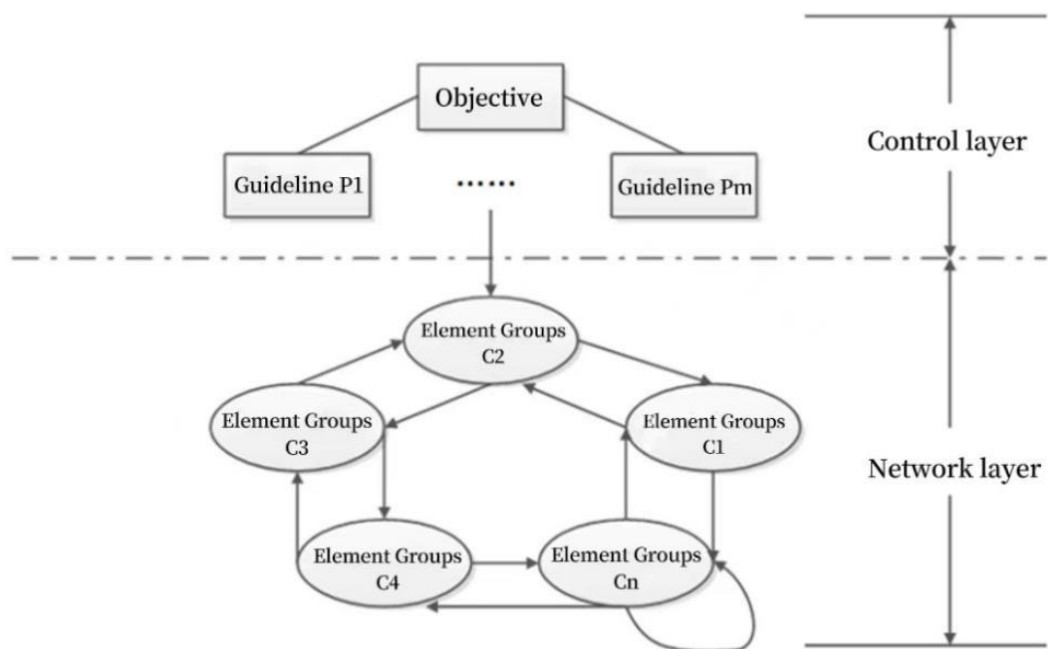

**Figure 1.** Structure diagram of network analysis method.

### 2.2.2. Network Hierarchy of Cargo Shipping Adaptability

To evaluate the impact of cargo shipping safety, it was necessary to apply the Delphi method to obtain the data required in the article, so 10 experts in cargo water (five scholars of shipping safety research and five experienced cargo ship captains) were invited to take part in the process of writing the article. The shipping safety researchers, three of whom were university professors in related fields and two of whom were managers of the Safety Division of the Shipping Administration, formed a group of experts to develop a network analysis method model. The purpose of this paper is to evaluate cargo shipping adaptability, so the criterion of subdivision layer in the index system is omitted, which has no influence on the calculation results, and the layer of influencing factors of cargo shipping is analyzed directly.

Through expert analysis, the interrelationship between the five factors affecting cargo shipping safety is judged one by one and the structure of cargo shipping safety risk index system is established as an ANP network hierarchy, as shown in Figure 2. The control layer in the figure includes problem objectives and decision criteria and this layer and the network layer, which is composed of factors governed by the control layer, are influenced by each other internally.

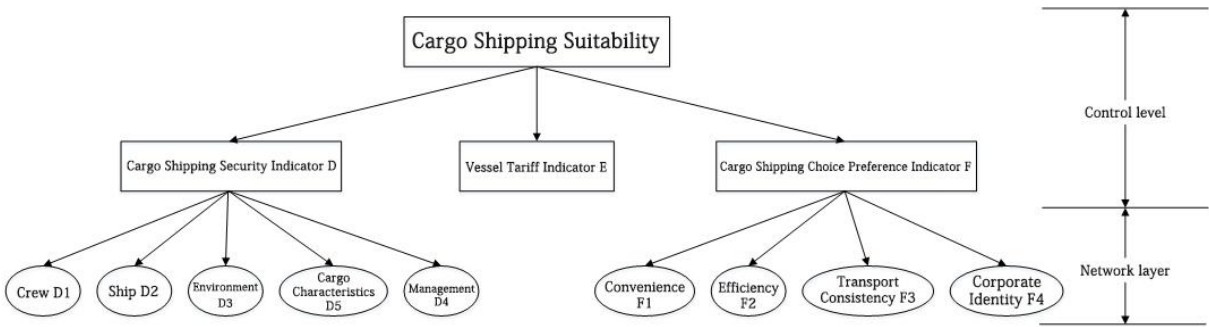

**Figure 2.** Hierarchy of the network of indicators for the evaluation of the adaptability of cargo shipping.

To establish an accurate ANP network model, it is necessary to clarify the interaction between the various factors. The network diagram of cargo shipping safety indicators and

the network diagram of cargo shipping choice preferences, as shown in Figures 3 and 4, were obtained in consultation with relevant experts.

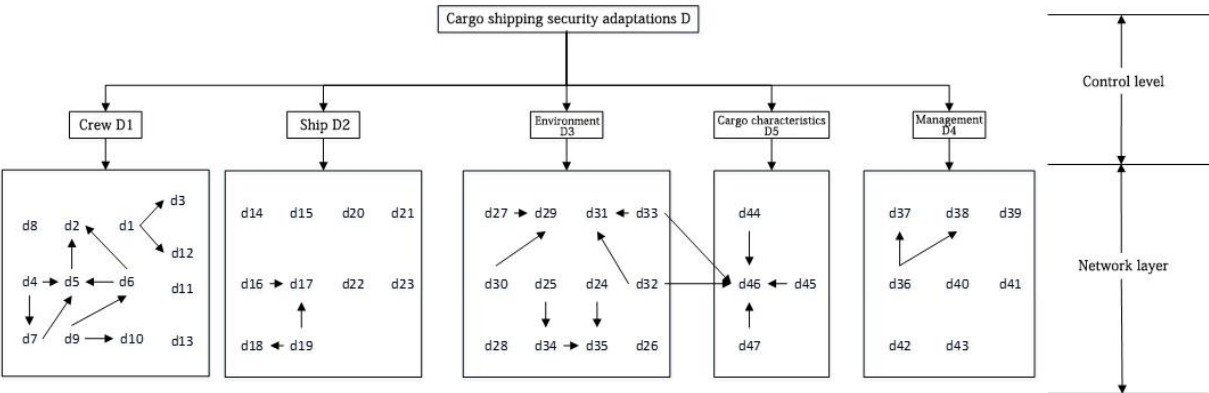

**Figure 3.** Network diagram of cargo shipping safety indicators.

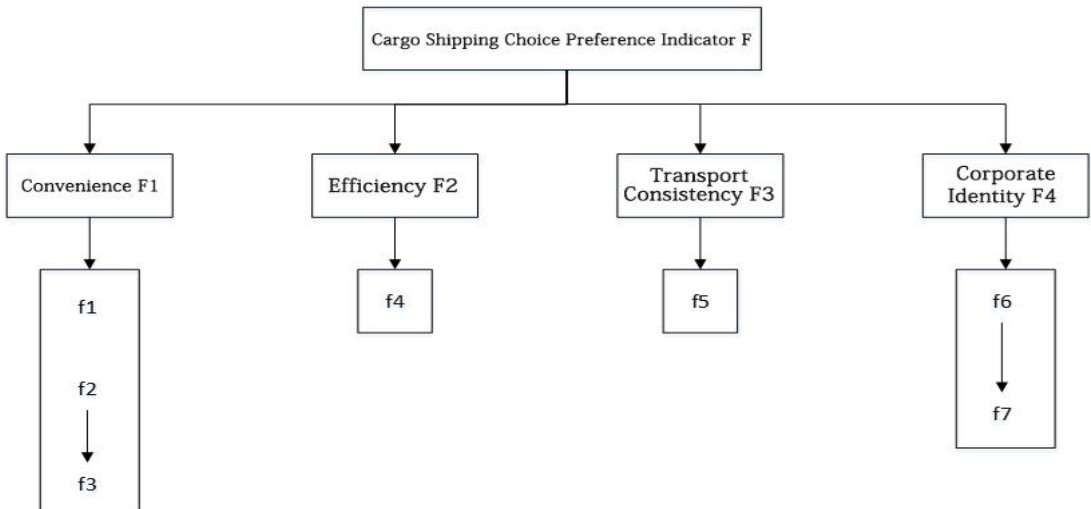

**Figure 4.** Network diagram of cargo shipping choice preference indicators.

### 2.2.3. Indicator Importance Judgement Matrix

Due to the large number of two-by-two judgment matrices involved in this paper, some representative matrices are listed for illustration. Using the 9-scalar method, experts were invited to make a two-by-two comparison of the first-level indicators of cargo shipping adaptability, and the two-by-two judgment matrix of the first-level indicators was obtained, as shown in Table 2.

**Table 2.** Two-by-two judgment matrix for first-level indicators.

| - | $D_1$ | $D_2$ | $D_3$ | $D_4$ | $D_5$ |
|---|---|---|---|---|---|
| $D_1$ | 1 | 1/2 | 1/4 | 1/7 | 3 |
| $D_2$ | 2 | 1 | 1/3 | 1/6 | 4 |
| $D_3$ | 4 | 3 | 1 | 1/4 | 6 |
| $D_4$ | 7 | 6 | 4 | 1 | 9 |
| $D_5$ | 1/3 | 1/4 | 1/6 | 1/9 | 1 |

Table 3 shows the secondary indicator judgment matrix, illustrated with $D_1$ as an example.

**Table 3.** Two-by-two judgment matrix of secondary indicators using $D_1$ as the judgment criterion.

| $D_1$ | $d_1$ | $d_2$ | $d_3$ | $d_4$ | $d_5$ | $d_6$ | $d_7$ | $d_8$ | $d_9$ | $d_{10}$ | $d_{11}$ | $d_{12}$ | $d_{13}$ |
|---|---|---|---|---|---|---|---|---|---|---|---|---|---|
| $d_1$ | 1 | 1/3 | 3 | 1/3 | 5 | 2 | 1/2 | 1/5 | 2 | 3 | 3 | 1 | 1/4 |
| $d_2$ | 3 | 1 | 5 | 1 | 7 | 4 | 2 | 1/3 | 4 | 5 | 5 | 3 | 1/2 |
| $d_3$ | 1/3 | 1/5 | 1 | 1/5 | 3 | 1/2 | 1/4 | 1/7 | 1/2 | 1 | 1 | 1/3 | 1/6 |
| $d_4$ | 3 | 1 | 5 | 1 | 7 | 4 | 2 | 1/3 | 4 | 5 | 5 | 3 | 1/2 |
| $d_5$ | 1/5 | 1/7 | 1/3 | 1/7 | 1 | 1/4 | 1/6 | 1/9 | 1/4 | 1/3 | 1/3 | 1/5 | 1/8 |
| $d_6$ | 1/2 | 1/4 | 2 | 1/4 | 4 | 1 | 1/3 | 1/6 | 1 | 2 | 2 | 1/2 | 1/5 |
| $d_7$ | 2 | 1/2 | 4 | 1/2 | 6 | 3 | 1 | 1/4 | 3 | 4 | 4 | 2 | 1/3 |
| $d_8$ | 5 | 3 | 7 | 3 | 9 | 6 | 4 | 1 | 6 | 7 | 7 | 5 | 2 |
| $d_9$ | 1/2 | 1/4 | 2 | 1/4 | 4 | 1 | 1/3 | 1/6 | 1 | 2 | 2 | 1/2 | 1/5 |
| $d_{10}$ | 1/3 | 1/5 | 1 | 1/5 | 3 | 1/2 | 1/4 | 1/7 | 1/2 | 1 | 1 | 1/3 | 1/6 |
| $d_{11}$ | 1/3 | 1/5 | 1 | 1/5 | 3 | 1/2 | 1/4 | 1/7 | 1/2 | 1 | 1 | 1/3 | 1/6 |
| $d_{12}$ | 1 | 1/3 | 3 | 1/3 | 5 | 2 | 1/2 | 1/5 | 2 | 3 | 3 | 1 | 1/4 |
| $d_{13}$ | 4 | 2 | 6 | 2 | 8 | 5 | 3 | 1/2 | 5 | 6 | 6 | 4 | 1 |

In addition to the comparative analysis of importance, a two-by-two judgment matrix of the degree of influence of two elements on the secondary criterion under the primary criterion was established. Table 4 shows the two-by-two judgment matrix with $d_9$ as the sub-criterion.

**Table 4.** Two-by-two judgment matrix for three-level indicators with $d_9$ as sub-criterion.

| $d_9$ | $d_6$ | $d_9$ | $d_{10}$ |
|---|---|---|---|
| $d_6$ | 1 | 1/4 | 1 |
| $d_9$ | 4 | 1 | 1/4 |
| $d_{10}$ | 1 | 4 | 1 |

2.2.4. Indicator Identification Calculation

A total of 55 influencing factors were involved in this paper, so Super Decisions was used to solve them. The network hierarchy was modeled in the Super Decisions software Super Decisions V2.10. The expert judgments were processed and input into the software. After the calculation by Super Decisions, the local dominance and global dominance of each factor were finally obtained, as shown in Table 5. The local dominance characterized the proportion of the weight of the factor in the set of primary indicators and the global dominance characterized the proportion of the weight of the factor in the security risk of cargo shipping.

**Table 5.** Local dominance and global dominance.

| Factor Name | Local Dominance Degree | Global Dominance Degree |
|---|---|---|
| Cargo shipping safety adaptability D | 0.637 | |
| Cargo shipping tariff adaptation E | 0.25828 | |
| Cargo shipping preference adaptability F | 0.10472 | |
| Crew $D_1$ | 0.23895 | 0.152211 |
| Ship $D_2$ | 0.15728 | 0.100187 |
| Environment $D_3$ | 0.07637 | 0.048648 |
| Management $D_4$ | 0.03187 | 0.020301 |
| Cargo characteristics $D_5$ | 0.49553 | 0.315653 |
| Convenience $F_1$ | 0.28879 | 0.030242 |
| Efficiency $F_2$ | 0.47619 | 0.049867 |

**Table 5.** *Cont.*

| Factor Name | Local Dominance Degree | Global Dominance Degree |
|---|---|---|
| Transport consistency $F_3$ | 0.17595 | 0.018425 |
| Corporate image $F_4$ | 0.05908 | 0.006187 |
| Responsibility $d_1$ | 0.03031 | 0.004614 |
| Psychological quality $d_2$ | 0.26047 | 0.039646 |
| Safety awareness $d_3$ | 0.09567 | 0.014562 |
| Academic qualifications $d_4$ | 0.01388 | 0.002113 |
| Operating ability $d_5$ | 0.23268 | 0.035416 |
| Length of service on board $d_6$ | 0.06947 | 0.010574 |
| Learning ability $d_7$ | 0.03147 | 0.00479 |
| Participation in training $d_8$ | 0.00749 | 0.00114 |
| Age $d_9$ | 0.04631 | 0.007049 |
| Health condition $d_{10}$ | 0.09458 | 0.014396 |
| Fatigue level $d_{11}$ | 0.07142 | 0.010871 |
| How much love for work $d_{12}$ | 0.03638 | 0.005537 |
| Satisfaction with treatment $d_{13}$ | 0.00987 | 0.001502 |
| Level of ship maintenance $d_{14}$ | 0.02915 | 0.00292 |
| Ship's age $d_{15}$ | 0.03035 | 0.003041 |
| Ship's tonnage $d_{16}$ | 0.01979 | 0.001983 |
| Structural stability of the ship $d_{17}$ | 0.15644 | 0.015673 |
| Hull strength $d_{18}$ | 0.13666 | 0.013692 |
| Cargo requirements for ship structure $d_{19}$ | 0.17995 | 0.018029 |
| Communication and signaling equipment $d_{20}$ | 0.07283 | 0.007297 |
| Lashing equipment $d_{21}$ | 0.07581 | 0.007595 |
| Fire-fighting system $d_{22}$ | 0.11908 | 0.01193 |
| Cargo requirements for ship equipment $d_{23}$ | 0.17995 | 0.018029 |
| Channel markings $d_{24}$ | 0.03421 | 0.001664 |
| Channel width $d_{25}$ | 0.02336 | 0.001136 |
| Excess water depth $d_{26}$ | 0.05003 | 0.002434 |
| Water speed $d_{27}$ | 0.01168 | 0.000568 |
| Current direction $d_{28}$ | 0.01168 | 0.000568 |
| Wave height $d_{29}$ | 0.2859 | 0.013908 |
| Wind $d_{30}$ | 0.07265 | 0.003534 |
| Visibility $d_{31}$ | 0.10935 | 0.00532 |
| Temperature $d_{32}$ | 0.10543 | 0.005129 |
| Air humidity $d_{33}$ | 0.15145 | 0.007368 |
| Vessel density $d_{34}$ | 0.04689 | 0.002281 |
| Waterway order $d_{35}$ | 0.09736 | 0.004736 |
| Soundness of safety management system $d_{36}$ | 0.08228 | 0.00167 |
| Degree of safety organization and training $d_{37}$ | 0.03786 | 0.000769 |
| Degree of safety publicity $d_{38}$ | 0.03786 | 0.000769 |
| Soundness and implementation of emergency rescue system $d_{39}$ | 0.0417 | 0.000847 |
| Degree of enforcement of safety laws and regulations $d_{40}$ | 0.2049 | 0.00416 |
| Degree of information of shipping safety $d_{41}$ | 0.0782 | 0.001588 |
| Degree of application of cargo stowage software $d_{42}$ | 0.25934 | 0.005265 |
| Soundness of cargo shipping management regulations $d_{43}$ | 0.25786 | 0.005235 |
| Degree of cargo shipping time safety $d_{44}$ | 0.03796 | 0.011982 |
| Degree of safety of loading position $d_{45}$ | 0.03796 | 0.011982 |
| Physical and chemical characteristics of cargoes $d_{46}$ | 0.69109 | 0.218144 |
| Degree of meeting the requirements of packaging $d_{47}$ | 0.23298 | 0.073541 |
| Vessel tariff $e_1$ | 1 | 0.25828 |
| Shipping interval $f_1$ | 0.25527 | 0.00772 |
| Ease of switching modes of transport $f_2$ | 0.60429 | 0.018275 |
| Convenience of port location $f_3$ | 0.14043 | 0.004247 |
| Speed of transport $f_4$ | 1 | 0.049867 |
| Transport consistency $f_5$ | 1 | 0.018425 |
| Enterprise size $f_6$ | 0.33333 | 0.002062 |
| Management standardization $f_7$ | 0.66667 | 0.004125 |

Through the analysis of the above results, we found that the dominance of certain factors was less than 5‰ and these factors had little influence on the adaptability of cargo shipping, so they were eliminated to simplify the calculation. The simplified indicators were re-arranged in descending order of global dominance: learning ability $d_7$, order of waterway $d_{35}$, responsibility $d_1$, the convenience of port location $f_3$, degree of enforcement of safety laws and regulations $d_{40}$, management standardization $f_7$, wind $d_{30}$, age condition of ship $d_{15}$, maintenance level of ship $d_{14}$, surplus water depth $d_{26}$, ship density $d_{34}$, education $d_4$, enterprise size $f_6$, tonnage condition of ship $d_{16}$, soundness of safety management system $d_{36}$, waterway marking $d_{24}$, level of shipping safety information $d_{41}$, satisfaction with treatment $d_{13}$, participation in training $d_8$, channel width $d_{25}$, soundness and implementation of emergency rescue system $d_{39}$, degree of safety organization and training $d_{37}$, degree of safety publicity $d_{38}$, current speed $d_{27}$ and current direction $d_{28}$. These 25 index factors were rounded off and the remaining 30 factors were renumbered. On this basis, the new local dominance and global dominance are shown in Table 6.

**Table 6.** Optimized cargo shipping adaptability evaluation indicators and dominance.

| Target Layer | System Layer | Guideline Layer | Code Breakdown | Factor Layer | Global Dominance Degree |
|---|---|---|---|---|---|
| Cargo shipping adaptability | Cargo shipping security adaptability D | Crew $D_1$ | Personality traits $Y_{11}$ | Psychological quality $y_1$ | 0.042144 |
| | | | | Safety awareness $y_2$ | 0.015479 |
| | | | Personal ability $Y_{12}$ | Operating ability $y_3$ | 0.037647 |
| | | | | Length of service on board $y_4$ | 0.01124 |
| | | | Physiological condition $Y_{13}$ | Age $y_5$ | 0.007493 |
| | | | | Health condition $y_6$ | 0.015303 |
| | | | | Fatigue level $y_7$ | 0.011556 |
| | | | Motivation $Y_{14}$ | How much love for work $y_8$ | 0.005886 |
| | | ShipD2 | Hull structure $Y_{21}$ | Structural stability of the ship $y_9$ | 0.01666 |
| | | | | Strength of ship's hull $y_{10}$ | 0.014555 |
| | | | | Cargo requirements for ship structure $y_{11}$ | 0.019165 |
| | | | Ship equipment $Y_{22}$ | Communication and signaling equipment $y_{12}$ | 0.007757 |
| | | | | Lashing equipment $y_{13}$ | 0.008073 |
| | | | | Fire-fighting system $y_{14}$ | 0.012682 |
| | | | | Cargo requirements for ship equipment $y_{15}$ | 0.019165 |

**Table 6.** *Cont.*

| Target Layer | System Layer | Guideline Layer | Code Breakdown | Factor Layer | Global Dominance Degree |
|---|---|---|---|---|---|
| | | Environment $D_3$ | Hydrographic conditions $Y_{31}$ | Wave height $y_{16}$ | 0.014784 |
| | | | Meteorological conditions $Y_{32}$ | Visibility $y_{17}$ | 0.005655 |
| | | | | Temperature $y_{18}$ | 0.007832 |
| | | | | Air humidity $y_{19}$ | 0.005452 |
| | | Management $D_4$ | Cargo stowage software application $Y_{41}$ | Degree of application of cargo stowage software $y_{20}$ | 0.005597 |
| | | | Cargo shipping regulations $Y_{42}$ | Degree of soundness of cargo shipping regulations $y_{21}$ | 0.005565 |
| | | Cargo characteristics $D_5$ | Cargo shipping time $Y_{51}$ | Degree of cargo shipping time security $y_{22}$ | 0.012737 |
| | | | Cargo loading location $Y_{52}$ | Degree of security of loading location $y_{23}$ | 0.012737 |
| | | | Characteristics of the cargo itself $Y_{53}$ | Physical and chemical characteristics of cargoes $y_{24}$ | 0.231886 |
| | | | Cargo packaging characteristics $Y_{54}$ | The degree of meeting the requirements of packaging $y_{25}$ | 0.078174 |
| | Cargo shipping tariff adaptability E | Shipping Prices $E_1$ | Vessel tariff $E_{11}$ | Vessel tariff $y_{26}$ | 0.274551 |
| | Cargo shipping options preferences adaptability F | Convenience $F_1$ | Shipping interval $F_{11}$ | Shipping interval $y_{27}$ | 0.008206 |
| | | | Ease of switching transport modes $F_{12}$ | Ease of switching modes of transport $y_{28}$ | 0.019426 |
| | | Efficiency $F_2$ | Speed of transportation $F_{21}$ | Speed of transport $y_{29}$ | 0.053008 |
| | | Shipping Consistency $F_3$ | Transport consistency $F_{31}$ | Transport consistency $y_{30}$ | 0.019586 |

## 3. Model Construction

### 3.1. Safety Adaptation Model Construction

Bayesian network, also known as confidence network, is an extension of Bayes' method and was proposed by Judea Pearl in 1988 as a probability-based uncertainty and multivariate inference type network. Suitable for representing and analyzing a variety of uncertain as well as probabilistic events, the network is applied to decisions that conditionally depend on multiple control factors and can make correct inferences from knowledge or information with low completeness, low accuracy, or less certainty [50].

Bayesian networks have been widely used in many fields such as fault diagnosis, data mining, medical diagnosis and traffic safety, employing their unique form of uncertain knowledge expression, rich probabilistic expression capability and incremental learning by synthesizing a priori knowledge. Especially in the field of traffic safety, Bayesian networks have been successfully applied to traffic disaster causation analysis, traffic safety warning and traffic safety evaluation [51].

The theoretical basis of Bayesian network inference is mainly the Bayesian formulation. The Bayesian formula is also called the posterior probability formula. Let the prior probability be $P(B_i)$, if $P(A_j|B_i)$ is known, where $i = 1, 2, \cdots, n$, $j = 1, 2, \cdots, m$. Then the posterior probability calculated by Bayesian formula is

$$P(B_i|A_j) = \frac{P(B_i)P(A_j|B_i)}{\sum_{k=1}^{m} P(B_i)P(A_k|B_i)} \tag{1}$$

The Bayesian formula is known and, according to the characteristics of Bayesian net inference, probabilistic analysis can be performed on any node in the Bayesian net. If the prior probability of a parent node is given, the posterior probability of a child node can be calculated; conversely, the posterior probability of a child node is known and the prior probability of a parent node can also be calculated. For example, suppose the posterior probability that node $E_j$ is observed to be in state $e_{j0}$ and node $E_i$ is in state $e_{io}$.

$$P(E_i = e_{io}|E_j = e_{j0}) = \sum_{\substack{E_1, \cdots E_{i-1}, E_{i+1}, \cdots \\ E_{j-1}, E_{j+1} \cdots E_N}} \frac{P(E_k = e_k, E_i = e_{i0}, E_j = e_{j0}, 1 \leq k \leq N, k \neq i, k \neq j)}{P(E_j = e_{j0})} \tag{2}$$

where $E_k(1 < i < N)$, corresponds to the nodes in the Bayesian network, N is the number of nodes in the Bayesian network, $e_k \in \Omega$ is used to characterize the state of node $E_k$ and $\Omega_k$ is the state space of node $E_k$.

### 3.1.1. Green Shipping Safety Bayesian Network Topology

According to the previous analysis, the 25 factors that have a significant impact on cargo shipping safety: psychological quality $y_1$, safety awareness $y_2$, operational ability $y_3$, length of service on board $y_4$, age $y_5$, health condition $y_6$, fatigue level $y_7$, love for work $y_8$, structural stability of the ship $y_9$, hull strength $y_{10}$, cargo requirements for ship structure $y_{11}$, communication and signaling equipment $y_{12}$, lashing equipment $y_{13}$, fire-fighting system $y_{14}$, cargo requirements for ship equipment $y_{15}$, wave height $y_{16}$, visibility $y_{17}$, temperature $y_{18}$, air humidity $y_{19}$, degree of application of cargo stowage software $y_{20}$, soundness of cargo shipping regulations $y_{21}$, the safety of cargo time $y_{22}$, safety of loading location $y_{23}$, physical and chemical characteristics of cargo $y_{24}$ and degree of packaging meeting requirements $y_{25}$. The topology of the cargo shipping safety evaluation Bayesian network model was established, as shown in Figure 5.

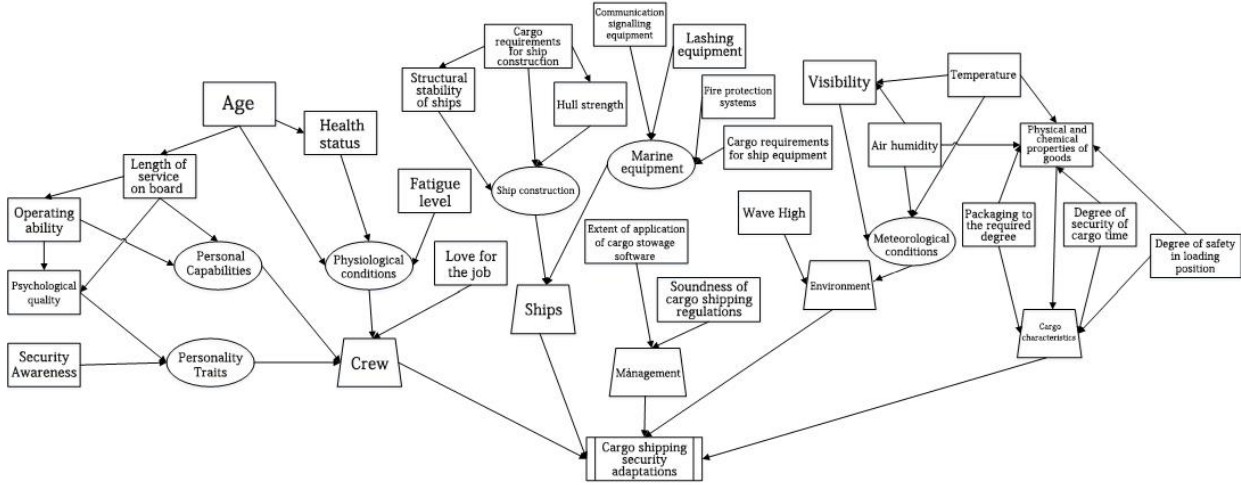

**Figure 5.** Cargo shipping safety evaluation Bayesian network topology.

### 3.1.2. Determination of the Value Domain of Cargo Shipping Safety Evidence Nodes

To quantify the specific impact of cargo shipping influence factors on shipping safety assessment, it was necessary to first divide the value domain of the nodal factors. The nodal value domain takes all integer values starting from 1, where 1 indicates good condition. A larger number represents worse condition. The various factors affecting the determination of evidence nodes for cargo shipping security are shown in the following tables.

### 3.1.3. Processing of Historical Data of Cargo Shipping Safety

The historical sample data mainly came from the data related to the risk of safety accidents of cargoes on 27 ships collected by a shipping company in 2012 and 2013, investigated by the author; the data on crew safety influence factors mainly correspond to the ships on which cargoes were loaded, among which the data of four factors, namely, psychological characteristics, safety awareness, operation ability and love for work, come from the monthly psychological and ability assessment of crew. The data on these four factors were obtained from the results of the software, while the age, age structure, health condition and fatigue level of the crew are obtained from their actual condition. The data on cargo characteristics' safety influencing factors came from the situation of the actual batch of cargo, which was obtained through the captain's record. Cargo shipping safety data contains 26 pieces of data information such as psychological quality, safety consciousness, operation ability, working age on board, age structure, etc. Since the collected historical data cannot be used directly for calculation, it was necessary to convert the statistical data into data that can be used directly according to the rules for defining nodal value domains from Tables 7–11, and the results are shown in Table 12.

**Table 7.** Table defining the value domain of the evidence node for crew factors affecting the safety of cargo shipping.

| Psychological Quality | Number of levels | 1 | 2 | 3 | |
|---|---|---|---|---|---|
| | Evaluation Criteria | $20 \leq y_1 \leq 30$ | $10 \leq y_1 < 20$ | $y_1 < 10$ | |
| Security Awareness | Number of levels | 1 | 2 | 3 | |
| | Evaluation Criteria | $85 \leq y_2 \leq 100$ | $60 \leq y_2 < 85$ | $y_2 < 60$ | |
| Operating Ability | Number of levels | 1 | 2 | 3 | |
| | Evaluation Criteria | $24 \leq y_3 \leq 30$ | $18 \leq y_3 < 24$ | $y_3 < 18$ | |
| Length of Service in the Ship | levels | 1 | 2 | 3 | |
| | Evaluation Criteria | $20\% \leq y_4$ | $20\% < y_4 \leq 40\%$ | $y_4 > 40\%$ | |
| Age Structure | Number of levels | 1 | 2 | 3 | 4 |
| | Evaluation Criteria | $y_5 > 7.2$ | $7.1 < y_5 \leq 7.2$ | $6.8 < y_5 \leq 7.1$ | $y_5 \leq 6.8$ |
| Health Status | levels | 1 | | 2 | |
| | Evaluation Criteria | Good and above | | Good or below | |
| Fatigue Level | Number of levels | 1 | 2 | 3 | |
| | Evaluation Criteria | $y_7 \leq 4$ | $4 < y_7 \leq 8$ | $y_7 > 8$ | |
| How Much Love for Work | levels | 1 | 2 | 3 | 4 |
| | Evaluation Criteria | $100 \leq y_8 \leq 124$ | $57 \leq y_8 \leq 99$ | $41 \leq y_8 \leq 56$ | $30 \leq y_8 \leq 40$ or $y_8 \geq 125$ |

**Table 8.** Table defining the value domain of the evidence node for cargo shipping safety ship factors.

| Ship Structural Stability | Number of levels | 1 | | 2 | |
|---|---|---|---|---|---|
| | Evaluation Criteria | $0.3 \leq y_9 \leq 1.2$ | | $y_9 < 0.3$ or $y_9 > 1.2$ | |
| Hull Strength | Number of levels | 1 | 2 | 3 | |
| | Evaluation Criteria | Higher than A/AH | Equal to A/AH | Lower than A/AH | |
| Cargo Requirements for Ship Structure | Number of levels | 1 | 2 | 3 | |
| | Evaluation Criteria | Very satisfying | Moderate satisfaction | Unsatisfied | |
| Communication Signal Equipment | Number of levels | 1 | | 2 | |
| | Evaluation Criteria | Installation | | Not installed | |
| Lashing Equipment | Number of levels | 1 | | 2 | |
| | Evaluation Criteria | $y_{13} \geq 120$ | | $y_{13} < 120$ | |
| Fire Fighting System | Number of levels | 1 | 2 | 3 | |
| | Evaluation Criteria | Very much in line with | Moderate compliance | Does not comply | |
| Cargo Requirements for Ship Equipment | Number of levels | 1 | 2 | 3 | |
| | Evaluation Criteria | Very satisfying | Moderate satisfaction | Unsatisfied | |

**Table 9.** Table defining the value domain of the evidence node for environmental factors for the safety of cargo shipping.

| Wave Height | Number of levels | 1 | 2 | 3 | |
|---|---|---|---|---|---|
| | Evaluation Criteria | $y_{16} \leq 1$ m | 1 m $< y_{16} \leq 2.5$ m | $y_{16} > 2.5$ m | |
| Visibility | Number of levels | 1 | 2 | 3 | 4 |
| | Evaluation Criteria | $y_{17} > 500$ m | 200 m $\leq y_{17} \leq 500$ m | 50 m $\leq y_{17} < 200$ m | $y_{17} < 50$ m |
| Temperature | Number of levels | 1 | 2 | 3 | |
| | Evaluation Criteria | Good | General | Bad | |
| Air Humidity | Number of levels | 1 | 2 | 3 | |
| | Evaluation Criteria | Good | General | Bad | |

**Table 10.** Table defining the value domain of the evidence node for cargo shipping safety management factors.

| Application of Cargo Stowage Software | Number of levels | 1 | 2 |
|---|---|---|---|
| | Evaluation Criteria | Applied | Not applied |
| Soundness of Cargo Shipping Regulations | Number of levels | 1 | 2 |
| | Evaluation Criteria | $y_{21} \geq 29$ | $y_{21} < 29$ |

**Table 11.** Table defining the value domain of the evidence node for the cargo security cargo characteristics factor for cargo shipping.

| Freight Time Security Level | Number of levels | 1 | 2 | |
|---|---|---|---|---|
| | Evaluation Criteria | Security | Insecure | |
| Safety Level of Loading Position | Number of levels | 1 | 2 | 3 |
| | Evaluation Criteria | Very safe | Moderately safe | Unsafe |
| Safety of Physical and Chemical Properties of Goods | Number of levels | 1 | 2 | 3 |
| | Evaluation Criteria | Very safe | Moderately safe | Unsafe |
| Packaging to the Required Degree | Number of levels | 1 | 2 | 3 |
| | Evaluation Criteria | Very safe | Moderately safe | Unsafe |

**Table 12.** Historical data processing results.

| Serial Number | $y_1$ | $y_2$ | $y_3$ | $y_4$ | $y_5$ | $y_6$ | $y_7$ | $y_8$ | $y_9$ | $y_{10}$ | $y_{11}$ | $y_{12}$ | $y_{13}$ | $y_{14}$ | $y_{15}$ | $y_{16}$ | $y_{17}$ | $y_{18}$ | $y_{19}$ | $y_{20}$ | $y_{21}$ | $y_{22}$ | $y_{23}$ | $y_{24}$ | $y_{25}$ | Cargo Shipping Security |
|---|---|---|---|---|---|---|---|---|---|---|---|---|---|---|---|---|---|---|---|---|---|---|---|---|---|---|
| 1 | 2 | 1 | 1 | 1 | 1 | 1 | 1 | 1 | 1 | 1 | 1 | 1 | 1 | 1 | 1 | 1 | 1 | 2 | 1 | 1 | 1 | 1 | 1 | 2 | 1 | 1 |
| 2 | 2 | 2 | 1 | 1 | 1 | 2 | 1 | 1 | 1 | 1 | 1 | 1 | 1 | 1 | 1 | 1 | 1 | 1 | 1 | 1 | 1 | 1 | 1 | 1 | 1 | 1 |
| 3 | 1 | 1 | 1 | 2 | 1 | 1 | 2 | 2 | 1 | 1 | 1 | 1 | 1 | 1 | 1 | 1 | 1 | 2 | 1 | 1 | 1 | 1 | 1 | 1 | 1 | 1 |
| 4 | 1 | 2 | 2 | 1 | 2 | 1 | 1 | 1 | 1 | 1 | 1 | 1 | 1 | 1 | 1 | 1 | 1 | 1 | 1 | 1 | 1 | 1 | 1 | 1 | 1 | 1 |
| 5 | 1 | 3 | 1 | 2 | 1 | 1 | 1 | 1 | 1 | 1 | 1 | 1 | 1 | 1 | 1 | 1 | 1 | 1 | 1 | 1 | 1 | 1 | 2 | 1 | 3 | 1 |
| 6 | 2 | 1 | 1 | 1 | 2 | 1 | 1 | 1 | 1 | 1 | 1 | 1 | 1 | 2 | 1 | 1 | 1 | 1 | 1 | 1 | 1 | 1 | 1 | 1 | 1 | 1 |
| 7 | 1 | 1 | 2 | 1 | 2 | 1 | 1 | 1 | 1 | 2 | 1 | 1 | 1 | 2 | 1 | 2 | 1 | 1 | 1 | 1 | 1 | 1 | 1 | 1 | 1 | 1 |
| 8 | 1 | 2 | 1 | 1 | 1 | 1 | 1 | 1 | 1 | 1 | 2 | 1 | 1 | 1 | 2 | 1 | 1 | 1 | 2 | 1 | 1 | 1 | 2 | 1 | 1 | 1 |
| 9 | 1 | 2 | 2 | 1 | 3 | 1 | 1 | 2 | 1 | 1 | 1 | 1 | 1 | 1 | 1 | 1 | 1 | 1 | 1 | 1 | 1 | 1 | 1 | 1 | 1 | 1 |
| 10 | 2 | 1 | 1 | 1 | 2 | 1 | 3 | 1 | 1 | 1 | 1 | 1 | 1 | 1 | 1 | 1 | 1 | 1 | 1 | 1 | 1 | 1 | 1 | 1 | 1 | 1 |
| 11 | 1 | 1 | 1 | 3 | 1 | 1 | 1 | 1 | 1 | 2 | 1 | 1 | 1 | 1 | 1 | 1 | 1 | 2 | 1 | 1 | 1 | 1 | 1 | 1 | 1 | 1 |
| 12 | 1 | 1 | 1 | 2 | 1 | 1 | 1 | 1 | 1 | 1 | 2 | 1 | 1 | 1 | 1 | 1 | 1 | 1 | 1 | 1 | 1 | 1 | 1 | 1 | 1 | 1 |
| 13 | 2 | 1 | 1 | 1 | 1 | 1 | 1 | 1 | 1 | 1 | 2 | 1 | 1 | 1 | 1 | 2 | 1 | 1 | 1 | 1 | 1 | 1 | 1 | 1 | 2 | 1 |
| 14 | 2 | 1 | 1 | 1 | 1 | 1 | 1 | 2 | 1 | 1 | 1 | 1 | 1 | 1 | 1 | 1 | 1 | 1 | 1 | 1 | 1 | 1 | 1 | 1 | 1 | 1 |
| 15 | 1 | 2 | 1 | 1 | 1 | 1 | 1 | 1 | 1 | 1 | 1 | 1 | 1 | 1 | 1 | 1 | 1 | 1 | 1 | 1 | 1 | 1 | 1 | 1 | 1 | 1 |
| 16 | 1 | 1 | 2 | 1 | 2 | 1 | 2 | 1 | 1 | 1 | 1 | 1 | 1 | 1 | 1 | 1 | 1 | 1 | 1 | 1 | 1 | 1 | 1 | 1 | 1 | 1 |
| 17 | 3 | 1 | 1 | 1 | 1 | 1 | 1 | 1 | 1 | 1 | 1 | 1 | 1 | 1 | 1 | 1 | 1 | 1 | 1 | 1 | 1 | 1 | 1 | 1 | 1 | 1 |
| 18 | 1 | 1 | 1 | 2 | 1 | 1 | 1 | 1 | 1 | 1 | 1 | 1 | 1 | 1 | 1 | 1 | 1 | 1 | 1 | 1 | 1 | 1 | 1 | 1 | 1 | 1 |
| . . . . . . | . . . | . . . | . . . | . . . | . . . | . . . | . . . | . . . | . . . | . . . | . . . | . . . | . . . | . . . | . . . | . . . | . . . | . . . | . . . | . . . | . . . | . . . | . . . | . . . | . . . | . . . . . . |
| 842 | 1 | 2 | 1 | 1 | 2 | 1 | 1 | 1 | 1 | 1 | 1 | 1 | 1 | 1 | 1 | 1 | 2 | 1 | 1 | 1 | 1 | 1 | 1 | 1 | 1 | 1 |
| 843 | 2 | 1 | 1 | 1 | 1 | 1 | 1 | 1 | 1 | 1 | 1 | 1 | 1 | 1 | 1 | 1 | 1 | 1 | 1 | 1 | 1 | 1 | 1 | 1 | 1 | 1 |

3.1.4. Application of GeNIe Software for Bayesian Network Inference for Cargo Shipping Security

The Bayesian network structure for cargo shipping security was established and the model was constructed in GeNIe software 5.0.0.0. The value domain of each evidence node was determined previously through Tables 7–11, and the processed data was input into the GeNIe software to obtain the results. From the cargo shipping safety evaluation results, it can be seen that the overall condition of cargo shipping is good: the probability of cargo shipping being safe is 0.71 and the probability of cargo shipping being dangerous is 0.29.

Thus, the formula for the degree of adaptation of cargo shipping safety is as follows:

$$HHA = \frac{D}{0.71} \tag{3}$$

where HHA is the cargo shipping safety adaptation degree and D is the cargo shipping safety probability.

### 3.2. Tariff Adaptability

The degree of adaptability of freight shipping rates reflects the degree to which the shipping rates are adapted to the object of transport. Usually, the shipping tariff is based on transportation costs and expected profit, as shipping can meet the requirements of cargo transportation accessibility, so a high or low tariff mainly depends on the value-added services provided by shipping suppliers. The higher the service level, the higher the corresponding tariff, i.e., the shipping tariff is proportional to the level of shipping service. Shipping vessels with higher tariffs can provide higher efficiency, safer storage and transportation conditions, and higher consistency of transportation for cargo.

The choice of shipping rates by cargo owners is mainly related to the characteristics of the cargo itself. Cargoes with a high value are more likely to be shipped at a higher rate, as their value is much greater than the freight rate, to ensure the safety of the cargo during transportation and to obtain faster service. In addition, if the goods themselves are special and have different transport requirements than ordinary goods, such as special storage conditions, a relatively strict time window, etc., it is possible to choose shipping suppliers with higher freight rates to obtain certain shipping services that are not offered by low-cost shipping.

Therefore, the following formula was used to calculate the cargo shipping tariff adaptability in this paper:

$$HYJ = 1 - \frac{E - E_{\min}}{E_{\max} - E_{\min}} \tag{4}$$

where *HYJ* the degree of adaptation of freight shipping tariffs, *E* is the actual tariff for the calculation period, *Emax* is the maximum tariff for the calculation period and *Emin* is the minimum tariff for the calculation period.

The value of the degree of adaptability of cargo shipping was between 0 and 1. The closer the value was to 1, the more adaptable the cargo shipping tariff to the object of transport, and the closer it was to 0, the less adaptable to the object of transport.

### 3.3. Choice Preference Adaptability

The paper simplified the cargo shipping choice preference indicators and found that the four indicators of departure interval $y_{27}$, ease of switching transport modes $y_{28}$, transport speed $y_{29}$ and transport consistency $y_{30}$ had a greater impact on cargo shipping adaptability. Graded scores were given to each of these four indicators to obtain Table 13.

The degree of adaptation of cargo shipping choice preferences can be derived from the following equation.

$$HXP = \frac{\sum_{j=1}^{4} f_j \beta_j}{10} \tag{5}$$

where $\beta_j$ is the local dominance of the cargo shipping choice preference indicator.

**Table 13.** Choice preference indicator grading score table.

| | Levels | 1 | 2 | 3 | 4 |
|---|---|---|---|---|---|
| **Shipping Interval** | Evaluation Criteria | $y_{27} \leq 1$ | $1 < y_{27} \leq 7$ | $7 < y_{27} \leq 15$ | $y_{27} > 15$ |
| | Score f1 | 10 | 8 | 6 | 4 |
| **Ease of Switching Transportation Modes** | Levels | 1 | 2 | 3 | |
| | Evaluation Criteria | 3 kinds | 2 kinds | 1 kinds | |
| | Score $f_2$ | 10 | 8 | 6 | |
| **Shipping Speed** | Levels | 1 | 2 | 3 | |
| | Evaluation Criteria | Quick | Medium | low | |
| | Score $f_3$ | 10 | 8 | 6 | |
| **Shipping Consistency** | Levels | 1 | 2 | 3 | 4 |
| | Evaluation Criteria | $y_{30} \geq 95\%$ | $80\% \leq y_{30} < 95\%$ | $60\% \leq y_{30} < 80\%$ | $y_{30} < 60\%$ |
| | Score $f_4$ | 10 | 8 | 6 | 4 |

*3.4. Cargo Shipping Suitability*

Cargo can be classified into three categories, A, B and C, based on cargo value and cargo characteristics, as shown in Table 14.

**Table 14.** Cargo classification table.

| Value of Goods Cargo Characteristics | High | Medium | Low |
|---|---|---|---|
| Bad | A | A | B |
| Medium | A | B | B |
| Good | B | C | C |

This article developed a questionnaire to investigate the degree of impact of three sub-adaptations affecting cargo shipping, based on three categories of goods, A, B and C. The impact probability table as shown in Table 15 is obtained.

**Table 15.** Table of impact probabilities for cargo shipping adaptability.

| Sub-Adaptation Cargo Classification | Cargo Shipping Security Adaptability | Cargo Shipping Tariff Adaptability | Cargo Shipping Choice Preference Adaptation |
|---|---|---|---|
| A | 0.85 | 0.03 | 0.12 |
| B | 0.79 | 0.11 | 0.10 |
| C | 0.70 | 0.25 | 0.05 |

This led to the formula for cargo shipping adaptability.

$$\begin{aligned} Category\ A: \ HHSA &= 0.85HHA + 0.03HYJ + 0.12HXP \\ Category\ B: \ HHSB &= 0.79HHA + 0.11HYJ + 0.10HXP \\ Category\ C: \ HHSC &= 0.70HHA + 0.25HYJ + 0.05HXP \end{aligned} \tag{6}$$

From the above three formulas, it was possible to calculate the cargo shipping suitability, which can be calculated by first classifying the transported cargo and calculating the cargo shipping suitability for each category of cargo.

**4. Discussion**

A ship belonging to a shipping company specializes in transporting zinc–aluminum alloy. The ship's freight rate for transporting zinc–aluminum alloy is CNY 25 per ton, while the highest freight rate for transporting zinc–aluminum alloy on the route is CNY 150 per

ton and the lowest is RMB 56 per ton. The shipping interval is 12 days, the transport speed is medium, and the sailing time is 8 days. The ship has a high transport consistency of approximately 0.84 and the transport connections at Dalian port are both road and rail.

Based on the above information, the cargo shipping safety suitability of this batch of zinc–aluminum alloy can be obtained from Equation (3) as 0.9311.

From Equation (5), we can obtain the cargo shipping rate adaptability of 1 for this batch of zinc–aluminum alloy.

From Equation (6), the cargo shipping preference suitability of this zinc-aluminum alloy batch is 0.7836.

For a batch of white sugar, because of its medium value and medium cargo characteristics, it is judged to be a class B cargo and its cargo shipping suitability is $HHS = 0.79HHA + 0.11HYJ + 0.10HXP = 0.9239$

According to the calculation results, the shipping suitability of this zinc–aluminum alloy shipment is 0.9239, which means that the ship is suitable for transporting it. The shipping suitability of the cargo can be used to select shipping suppliers and different tariffs and is a useful reference for selecting the appropriate ship for the transport object.

## 5. System Design

### 5.1. Purpose of Development

This paper established a cargo shipping service adaptability evaluation index system and applied the network analysis method and Bayesian network to construct the calculation method of cargo shipping service adaptability. However, to apply the evaluation method of shipping service adaptability in actual production and provide a reference basis for the selection of shipping service providers, it is necessary to obtain the specific value of shipping service adaptability in real-time and analyze the calculation results. Based on this requirement, this paper designed and developed the shipping service adaptability evaluation system.

### 5.2. System Components

The cargo shipping service adaptability evaluation system consists of multiple computers, communication equipment, and software, mainly including the following aspects.

(1) Various types of computers including servers and clients. The main role of the server is to provide services for other computers on the network; the client machine processes data and outputs results by accepting services from the server.

(2) Network adapters. Provide the structure for communication networks to connect to computers.

(3) Network transmission medium. Mainly includes network communication and interconnection equipment

(4) External devices. Includes all external hardware used by the network as a whole, such as printers, etc.

(5) Network software. This includes all the network resources that serve the system, including the cargo accumulation software.

The main purpose of the system designed in this paper was to calculate the cargo shipping adaptation and to analyze the various sub-adaptations that affect the shipping adaptation.

### 5.3. System Design

The cargo shipping service adaptability evaluation system designed in this paper mainly includes a user management module, a basic data module, a node data input module, and an adaptability calculation module, the specific structure of which is shown in Figure 6.

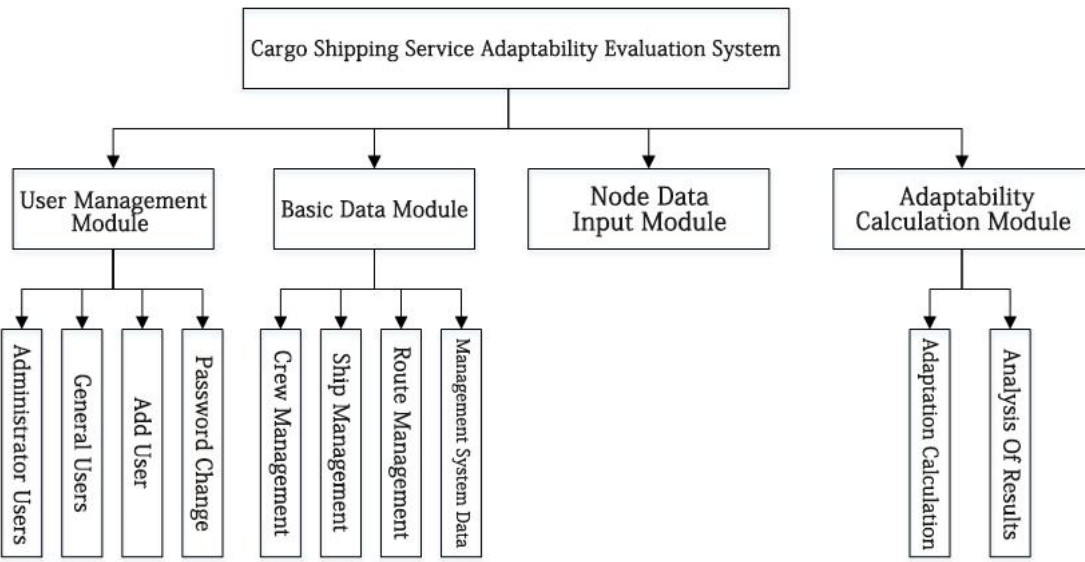

**Figure 6.** System architecture diagram.

The cargo shipping service adaptations are obtained and the main influencing factors of cargo shipping adaptations are determined by analyzing the sub-adaptations. It is also possible to view the proportion of each influencing factor of the cargo shipping security adaptation. The interface of the calculation results is shown in Figures 7 and 8.

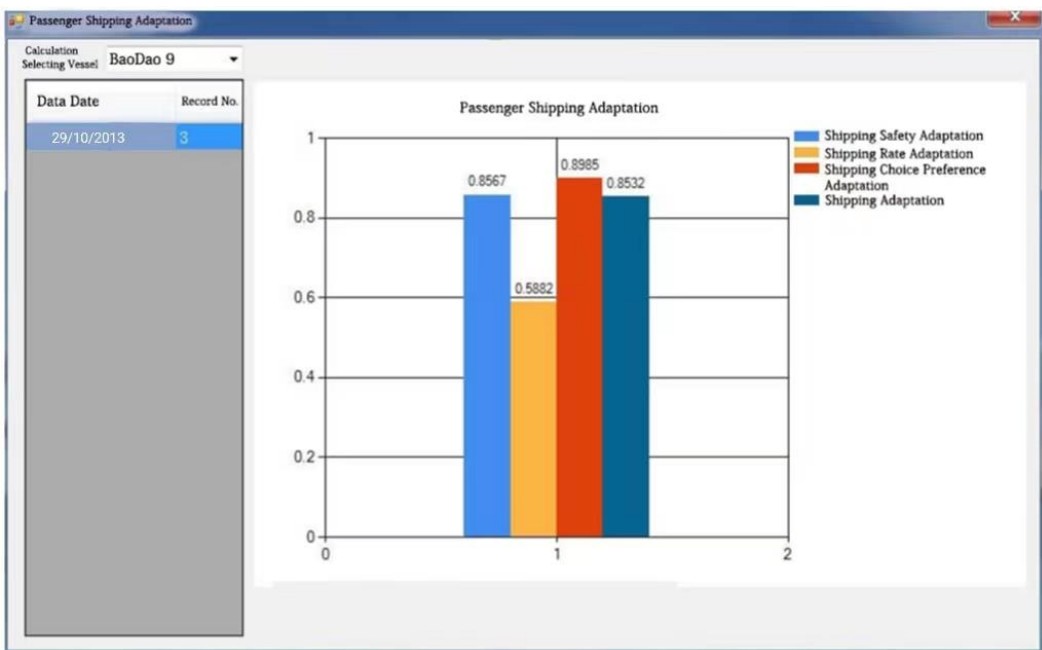

**Figure 7.** Adaptation calculation interface.

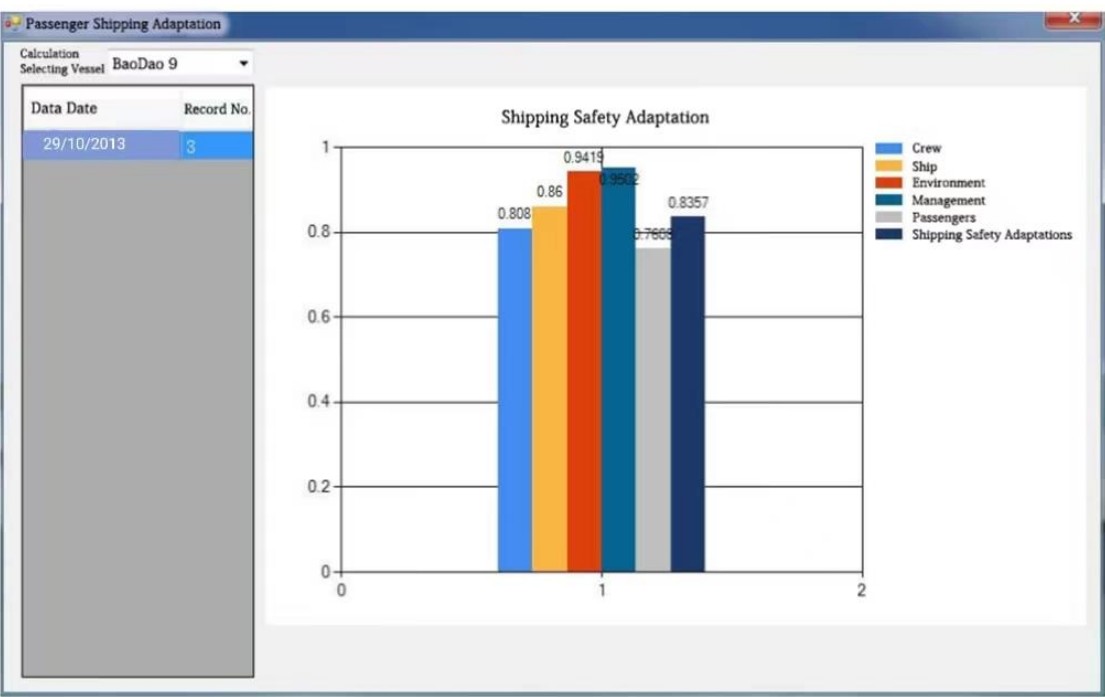

**Figure 8.** Security adaptation view screen.

## 6. Conclusions

In today's rapid economic development, water transportation, as an important transportation mode, takes up a large amount of capacity in the transportation of goods. In this paper, through the summary analysis of green shipping, as well as sustainability related research:

(1) Theoretically, by defining shipping service adaptability from the concept of biological adaptability, the analysis found that shipping adaptability of transportation objects mainly depends on the three aspects of shipping safety, shipping price and shipping choice preference and, by consulting experts in shipping, the influence index of shipping service adaptability was established; the network analysis method was used to simplify the index and the index of factors with greater influence on shipping service adaptability was obtained.

(2) In terms of method, based on the analysis of green shipping and cargo characteristics, the concept and calculation method of green cargo shipping service adaptability are constructed by applying ANP and Bayesian network models, establishing cargo shipping safety Bayesian network topology and obtaining cargo shipping safety adaptability through GeNIe software 5.0.0.0 simulation, which provides a basis for green shipping to measure and evaluate the degree of cargo adaptability. This paper designs and develops the evaluation system of shipping service adaptability, determines the functional modules of the system and designs the interface of the system, so that it can successfully apply the evaluation method of shipping service adaptability in actual production and provide a reference basis for the selection of green shipping service providers.

(3) In practice, through practical arithmetic analysis, it is proven that cargo shipping adaptability can be used for selecting shipping suppliers and different tariff selection, and is an effective reference standard when transporting objects to select appropriate ships. In this paper, by introducing the concept of shipping service adaptability and constructing a model of cargo shipping adaptability, the adaptation of shipping service to the transport object is evaluated.

(4) As a limitation of the research, research into shipping service adaptability is research involving all transportation objects. The article considers the characteristics of all shipping passengers and shipping cargo as much as possible, and improves the calculation method

of Bayesian net to construct the calculation method of shipping service adaptability. On this basis, software for shipping service adaptability measurement is developed, so that the calculation of shipping service adaptability can be directly applied to the selection of shipping supply ships for customers. However, due to the large research scope of this paper, it is difficult to be comprehensive, and the lack of data collection in the evaluation process leads to the existence of a certain subjectivity in the calculation results, thus making the evaluation difficult to be comprehensive and objective. Therefore, in future research, a more objective evaluation analysis of cargo shipping safety can be carried out by fuzzy-accurate Bayesian network and, for the problem of insufficient historical data, more objective data should be collected for calculation, in order to achieve more objective and practical results.

The core idea of green shipping is to realize the coordination of production activities of shipping enterprises with social and ecological benefits, to realize sustainable development, and to form a relatively competitive advantage over competitors, so as to obtain development among fierce competition. Sustainable development will be the inevitable route of future shipping development.

**Author Contributions:** Conceptualization, S.G.; methodology, S.G.; software, S.G.; validation, S.G. and W.N.; formal analysis, S.G.; investigation, S.G.; resources, W.N.; data curation, S.G.; writing—original draft preparation, S.G.; writing—review and editing, F.Z.; visualization, F.Z.; supervision, F.Z. and D.W. All authors have read and agreed to the published version of the manuscript.

**Funding:** 1. An Empirical Study on the Innovation of Spatial Social Governance Mechanism in the Revitalization of Northeast China and Its Integration into the New Development Pattern. General projects of the National Social Science Fund (Item No: 21BSH147). 2. Research on Key Issues of Big Data Social Governance in Jilin Province. Social Science Research Planning Major Project of Jilin Provincial Department of Education (JJKH20211303SK).

**Institutional Review Board Statement:** Not applicable.

**Informed Consent Statement:** Not applicable.

**Data Availability Statement:** No new data were created or analyzed in this study. Data sharing is not applicable to this article.

**Conflicts of Interest:** The authors declare no conflict of interest.

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
