# Peer review of "Optimization of Cargo Shipping Adaptability Modeling Evaluation Based on Bayesian Network Algorithm"

_sustainability, doi:10.3390/su141912856_

Round 1
Reviewer 1 Report
Dear Authors
This is an important topic for sustainable development in the area of ​​cargo shipping services. This article proposes the cargo shipping service adaptability index evaluation system with the aid of the network analysis method and Bayesian network to construct the calculation method of cargo shipping service adaptability. GeNIe software was used to model the network and obtain the results. An example of use was presented in Chapter 4.
There are only small details to review:
1) Improve the abstract, including more information in the last sentence or paragraph.
2) There is no space between the references and the words in the text. Line 32, 41, 42, and others, check the entire text.
3) Table 5 line 171, what is the column with the global domain? In both columns, it says Local Dominance, control. Still in relation to table 5, what does the indicator "Love for work" imply in this question? explain the relationship between the term and the search.
4) Line 184, Table 6 how to leave space left, the full word dominance is better, no need to separate the 2 basic columns of table 6.
5) Line 208 Figure 4, Increase the size of Figure 4 as much as possible
6) In table 7, the last line "Love for work", explains the relationship between the term and the search. Also in the Table 5 Evaluation Criteria row, the last column has a symbol before y8 that it is not possible to understand, please, check.
7) In table 8, row 1, "Ship Strutuctal Stability", "Evaluation criteria" column 2, has a symbol after y9 which cannot be understood, please, check.
8) Line 237, the probability of cargo transport being dangerous is 0.29 isn't it high? Explain this and value it properly in relation to other research, or articles.
9) Improve the sentence from line 363 to line 366, it is confusing.
Author Response
Response to Reviewer 1 Comments
Point 1: Improve the abstract, including more information in the last sentence or paragraph.
Response 1: For the issues mentioned in the abstract section, we have revised the abstract as a whole, going out of the large amount of redundant terminology and explaining that green transportation of goods is relevant, explaining why security, freight price and choice preference adaptation are included as variables under study. The model and algorithm have been explained and the novelty of the paper and the main contributions of the study have been added.
Point 2: There is no space between the references and the words in the text. Line 32, 41, 42, and others, check the entire text.
Response 2: In response to the question, we have checked all references throughout the text and have added spaces between all references and the text. This includes lines 32, 41, 42, and the full text.
Point 3: Table 5 line 171, what is the column with the global domain? In both columns, it says Local Dominance, control. Still in relation to table 5, what does the indicator "Love for work" imply in this question? explain the relationship between the term and the search.
Response 3: For this issue, it is indeed an oversight on our part that one of the columns is local dominance and the other is global dominance, which are not the same, but we wrote both as local dominance, and we have made changes to this. The local dominance characterizes the proportion of weight of factors in the set of first-level indicators, and the global dominance characterizes the proportion of weight of factors in the risk of cargo shipping security. For the indicator of “love for work”, perhaps we did not express it accurately enough, but it implies that it is one of the indicators for how much the respondents love their work.
Point 4: Line 184, Table 6 how to leave space left, the full word dominance is better, no need to separate the 2 basic columns of table 6.
Response 4: For the mentioned row 184, the blank in table 6 has been removed. Regarding the question that there is no need to separate the two columns in Table 6, after careful consideration, we believe that the overall trend of local dominance and global dominance is the same in Table 6, so we have adopted your suggestion to delete the local dominance and keep only the column of global dominance.
Point 5: Line 208 Figure 4, Increase the size of Figure 4 as much as possible
Response 5: For image 4 in row 208, its size has been increased.
Point 6: In table 7, the last line "Love for work", explains the relationship between the term and the search. Also in the Table 5 Evaluation Criteria row, the last column has a symbol before y8 that it is not possible to understand, please, check.
Response 6: The last line in Table 7, Love of work, may be a misunderstanding of our English expression, it expressed the meaning of the survey, how much the respondents love their work. for the work they do, which is one of the indicators. In the Table 5, there is no y8 indicator, perhaps the issue you raised was in Table 7, which we have identified and corrected.
Point 7: In table 8, row 1, "Ship Strutuctal Stability", "Evaluation criteria" column 2, has a symbol after y9 which cannot be understood, please, check.
Response 7: We have found the system in Table 8 and corrected it.
Point 8: Line 237, the probability of cargo transport being dangerous is 0.29 isn't it high? Explain this and value it properly in relation to other research, or articles.
Response 8: For your question about the probability of transporting dangerous goods in line 237 is 0.29 is not very high, we have carefully reviewed the relevant research references, (has been added in the reference section later) we give the explanation we studied is the historical data from ten years ago, so using the current calculation standards to see the data from ten years ago, although there are certain operational pitfalls, but not does not necessarily mean that there will be safety incidents. Because the initial standard was set high, the actual shipping data was found to be higher than the calculated value in the subsequent calculations, which was only a theoretical result of the initial study based on the preparation of the reference literature, and the risk of the data theory was relatively high, but the actual later calculations did not have such a high risk factor.
Point 9: Improve the sentence from line 363 to line 366, it is confusing.
Response 9: For the mentioned confusion of lines 363 to 366, we have modified this sentence.

Reviewer 2 Report
As a whole, the essay seems to have an excellent writing performance. However, there are some common errors. Therefore, the authors recommend that the paper be re-examined before submission to the journal.
(a) I suggest the authors revise the abstract appropriately.
(b) In the introduction section, the author has a relatively good performance, but the author needs to reinforce the importance of the topic green transportation under study.
(c) Please check the syntax and content of line 114-121.
(d) The methods used in the article are novel and accurately applied in terms of the methods used and proven from the literature. It is desirable to cite recent relevant studies on similar topics green transportation.
(e) From a general point of view, the authors should enhance the conclusion.
(f) I hope the author will strengthen the writing and grammar of this article.
Author Response
Response to Reviewer 2 Comments
Point 1: I suggest the authors revise the abstract appropriately.
Response 1: For the issues mentioned in the abstract section, we have revised the abstract as a whole, going out of the large amount of redundant terminology and explaining that green transportation of goods is relevant, explaining why security, freight price and choice preference adaptation are included as variables under study. The model and algorithm have been explained and the novelty of the paper and the main contributions of the study have been added.
Point 2: In the introduction section, the author has a relatively good performance, but the author needs to reinforce the importance of the topic green transportation under study.
Response 2: In response to your point about the importance of strengthening the topic under study with respect to green transportation, we have, after due consideration, reintroduced five new documents related to the topic of green transportation into the article and have highlighted them in red in the text.
Point 3: Please check the syntax and content of line 114-121.
Response 3: We have rechecked the writing and grammar of lines 114-121 and have highlighted them in red in the text.
Point 4: The methods used in the article are novel and accurately applied in terms of the methods used and proven from the literature. It is desirable to cite recent relevant studies on similar topics green transportation.
Response 4: After due consideration, reintroduced five new documents related to the topic of green transportation into the article and have highlighted them in red in the text.
Point 5: From a general point of view, the authors should enhance the conclusion.
Response 5: For the conclusion section, we completely revised the conclusion as a whole and made a point-by-point presentation of the theoretical, methodological, and practical aspects of the study, as well as the limitations and perspectives of the study, which also included the main significance of my research model and why it is better than other models.
Point 6: I hope the author will strengthen the writing and grammar of this article.
Response 6: Our team apologizes for the irregular language in the manuscript. We worked on the manuscript for a long time and repeatedly adding and removing sentences and sections apparently led to poor readability. We have now worked on both language and readability and have made improvements to the language and writing in the "English Editing" module of the MDPI journal system. After receiving feedback on the changes from the journal system, our team has considered your suggestions and made positive corrections to the language. And for all the revised parts we have highlighted them in red font and submitted the revised version of the document in the system. We truly hope to make substantial improvements in the process and language level.

Reviewer 3 Report
The paper aims to construct the calculation method of cargo green shipping safety, freight price, and choice preference adaptability are constructed, analysing historical data.
I recommend conducting some major reviews to continue the analysis of the paper.
1) Abstract: There are a lot of redundancies of terms in the abstract, it makes the abstract confusing and annoying. Remember that most readers will read your full paper if they are interested in what you wrote in the abstract. So it needs to be attractive. So, I suggest restructuring the abstract by initiating with a sentence explaining why cargo green shipping is relevant and why to study the variables related to safety, freight price, and choice preference adaptability. Then, you need to explain the source of the historical data (is it public?) and the method. I understand that you used modelling, so explain it considering that most readers are lay people. You should also explain what is the novelty of your paper and the main contributions of your study.
2) Introduction - The introduction should contain the contextualization of the topic exploited (this is ok), the discussion of the research gap (I didn´t find it), citing also why is relevant to study this specific problem related to green cargo shipping, and how do you advance related to the previously published studies. Then, you will present your RQ and your objective, explicitly. After this, you need to include a sentence related to the method used and data collection and analysis. moreover, the novelty of the paper should be addressed as the contributions (theoretical, methodological and practical).
3. Literature review - there is no discussion related to the state-of-art of the topic addressed. The sections included in the literature review are very succinct, which makes it difficult for lay readers to understand the paper. It is necessary also to point out the main papers dealing with similar topics to compare in which way your study differentiates from these. In the analysis of the results you will need to compare your results with the results of these papers.
4) Methods - You need to explain in this section what method and model you used to analyse the data. You did use mathematical programming, so you need to make it clear by following the patterns of scientifical communication. You should explain that this is a quantitative approach, and explain what is the type and source of data. The database is public or classified? which is the period of data? table 1 is too large and is not attractive, please try to condense and well present it. Please explain how the 10 experts in cargo shipping were selected. Which were the criteria to choose them, and what is the characterization of their profiles? You did use an ANP network model, which is from the Multicriteria approach, I did not find in any part of the paper an explanation of what is MCDA approach and the ANP method, nor, why did you choose ANP instead of other methods? the ANP is appropriate when you have a group decision, with the opinion of multiple stakeholders, how did you aggregate the opinions of 10 experts?
5) Results - the results should be more discussed in light of the theory and the previous published studies. You did present a lot of results, since ANP results and then the Bayesian model, but I ask you: how your study advances considering the previous studies related to this topic? As you did not cite the state of art in the literature review, it is hard to evaluate, so you need to address these elements. I suggest including a section of discussion of the results, comparing with the literature and inserting some figures or tables synthesising the main results and addressing the main implications to adopt the model proposed.
6) Conclusion: You should address the limitations of your study and the main contributions in terms of theoretical, methodological and practical contributions. What are the main implications of the adoption of your model? why is it better than the others proposed?
7) there are several typos and some English grammar errors. Please analyse and review carefully.
8) Please insert literature related to MCDA, ANP and state of art related to green cargo shipping.
9) I also found problems in the format considering the sustainability template.
Author Response
Response to Reviewer 3 Comments
Point 1: Abstract: There are a lot of redundancies of terms in the abstract, it makes the abstract confusing and annoying. Remember that most readers will read your full paper if they are interested in what you wrote in the abstract. So it needs to be attractive. So, I suggest restructuring the abstract by initiating with a sentence explaining why cargo green shipping is relevant and why to study the variables related to safety, freight price, and choice preference adaptability. Then, you need to explain the source of the historical data (is it public?) and the method. I understand that you used modelling, so explain it considering that most readers are lay people. You should also explain what is the novelty of your paper and the main contributions of your study.
Response 1: For the issues mentioned in the abstract section, we have revised the abstract as a whole, going out of the large amount of redundant terminology and explaining that green transportation of goods is relevant, explaining why security, freight price and choice preference adaptation are included as variables under study. The model and algorithm have been explained and the novelty of the paper and the main contributions of the study have been added.
Point 2: Introduction - The introduction should contain the contextualization of the topic exploited (this is ok), the discussion of the research gap (I didn´t find it), citing also why is relevant to study this specific problem related to green cargo shipping, and how do you advance related to the previously published studies. Then, you will present your RQ and your objective, explicitly. After this, you need to include a sentence related to the method used and data collection and analysis. moreover, the novelty of the paper should be addressed as the contributions (theoretical, methodological and practical).
Response 2:
- In terms of research gaps, we found through a careful study of the literature that research in shipping has focused mainly on the two directions of transport safety and satisfaction, while there is little research on the adaptability of shipping services. This is the entry point of the research gap between us and our predecessors.
- In the study of green cargo transportation, we mentioned the relevant green cargo transportation part, and we re-cited two references related to green cargo transportation in the manuscript for discussion.
- After the proposed research questions and objectives, we added a sentence related to the methodology used and data collection and analysis based on your suggestions. It is also marked in red font.
- In terms of the novelty of the paper, we have reorganized and revised this section, dealing with it according to theoretical, methodological, and practical aspects.
Point 3: Literature review - there is no discussion related to the state-of-art of the topic addressed. The sections included in the literature review are very succinct, which makes it difficult for lay readers to understand the paper. It is necessary also to point out the main papers dealing with similar topics to compare in which way your study differentiates from these. In the analysis of the results you will need to compare your results with the results of these papers.
Response 3: In the literature review section, we have rearranged and added a lot of relevant literature in order to avoid difficulties for non-specialist readers to understand the paper, and the applied methods include Bayesian networks as well as network analysis methods. We also add a discussion of the literature and a comparison with this paper at the end of each section. The relevant changes have been highlighted in red in the text.
Point 4: Methods - You need to explain in this section what method and model you used to analyse the data. You did use mathematical programming, so you need to make it clear by following the patterns of scientifical communication. You should explain that this is a quantitative approach, and explain what is the type and source of data. The database is public or classified? which is the period of data? table 1 is too large and is not attractive, please try to condense and well present it. Please explain how the 10 experts in cargo shipping were selected. Which were the criteria to choose them, and what is the characterization of their profiles? You did use an ANP network model, which is from the Multicriteria approach, I did not find in any part of the paper an explanation of what is MCDA approach and the ANP method, nor, why did you choose ANP instead of other methods? the ANP is appropriate when you have a group decision, with the opinion of multiple stakeholders, how did you aggregate the opinions of 10 experts?
Response 4:
- In order to evaluate the impact of cargo shipping safety, our team took a comprehensive approach and finally applied the Delphi method to obtain the required data in the article, and invited a group of 10 experts in cargo water to score the article during the writing process. In order to get the influence of the state in which each factor in the article is located on cargo safety and to clarify the value domain of each node, the author invited relevant experts to have a discussion and conducted a questionnaire survey to the management and senior senior captains of shipping companies, so as to get the corresponding data, and through statistical analysis, we finally got the grading of each node. For cargo shipping history data, it comes from the data related to the risk of safety accidents of cargo on 27 ships collected by a shipping company in 2012 and 2013 investigated by the team.
- Regarding your question that Table 1 is too large and unattractive, after careful consideration, we delete the first column of the target layer in Form 1, so that on the whole, there are only five layers in the table: system layer, criterion layer, criterion subdivision, and factor layer in the cargo shipping evaluation index system, which specifically include the shadow 55 influencing factors of cargo shipping adaptability.
- To evaluate the impact of cargo shipping safety, it was necessary to apply the Delphi method to obtain the data required in the article, so 10 experts in cargo water (5 scholars of shipping safety research and 5 experienced cargo ship captains) were invited in the process of writing the article. Three of the shipping safety researchers were university professors in the relevant field and two were managers of the Safety Division of the Shipping Administration to form a group of experts to develop the network analysis method model. The answer to your question about how to choose ten experts in cargo transportation, what are the criteria for choosing them and what are their characteristics, we give is that the region where the authors study is inland not coastal, inland water transport is not developed and accounts for a very small percentage, so we invite professional scientific researchers from relevant research universities and relevant transportation departments mainly for the more influential 10 experts in the field of water transport.
- 4.In order to make the paper better understood by non-specialist readers, we have added the basic principles of network analysis method as well as diagrams in the second part of the paper to better explain the MCDA method and ANP method.
Point 5: Results - the results should be more discussed in light of the theory and the previous published studies. You did present a lot of results, since ANP results and then the Bayesian model, but I ask you: how your study advances considering the previous studies related to this topic? As you did not cite the state of art in the literature review, it is hard to evaluate, so you need to address these elements. I suggest including a section of discussion of the results, comparing with the literature and inserting some figures or tables synthesising the main results and addressing the main implications to adopt the model proposed.
Response 5:
- In the part of our results, by establishing the formulae for calculating three adaptations of shipping safety adaptations of transport objects, freight rate adaptations of transport objects and preference adaptations of transport objects in the previous part, and classifying the corresponding transport objects and distributing questionnaires for different categories respectively, the survey obtained the construction method of shipping adaptations of different categories of transport objects. The method was applied to the practical study, and the usability and validity of the method were proved. And in the system design part, the cargo shipping service adaptability evaluation index system established in this paper, the network analysis method and Bayesian network are applied to construct the calculation method of cargo shipping service adaptability, this paper designs and develops the shipping service adaptability evaluation system, gets the specific value of shipping service adaptability in real time and analyzes the calculation results, which makes it possible to apply shipping service adaptability in practical production This evaluation method provides a reference basis for the selection of shipping service providers. We believe that this is one of our conclusions and research advances, as well as solving the main implications of adopting the proposed model.
- In response to your question about the lack of references to the state of the art in the literature review, we have reworked the technology and methods section of the references to include references to previous literature on methods and added an explanation of the basic principles of the network analysis method used in the article in Chapter 2 andchapter 3, we added the basic principles of Bayesian networks and the introduction of Bayesian networks in the model construction section.
Point 6: Conclusion: You should address the limitations of your study and the main contributions in terms of theoretical, methodological and practical contributions. What are the main implications of the adoption of your model? why is it better than the others proposed?
Response 6: For the conclusion section, we completely revised the conclusion as a whole and made a point-by-point presentation of the theoretical, methodological, and practical aspects of the study, as well as the limitations and perspectives of the study, which also included the main significance of my research model and why it is better than other models.
Point 7: there are several typos and some English grammar errors. Please analyse and review carefully.
Response 7: Our team apologizes for the irregular language in the manuscript. We worked on the manuscript for a long time and repeatedly adding and removing sentences and sections apparently led to poor readability. We have now worked on both language and readability and have made improvements to the language and writing in the "English Editing" module of the MDPI journal system. After receiving feedback on the changes from the journal system, our team has considered your suggestions and made positive corrections to the language. And for all the revised parts we have highlighted them in red font and submitted the revised version of the document in the system. We truly hope to make substantial improvements in the process and language level.
Point 8: Please insert literature related to MCDA, ANP and state of art related to green cargo shipping.
Response 8: We have researched and added literature on the methodology and the state of the art related to green cargo transportation in the literature review section as well as in the second section, respectively.
Point 9: I also found problems in the format considering the sustainability template.
Response 9: In response to your question “I also found problems in the format considering the sustainability template.”, our team has made changes to the entire article and its formatting, and has used the English Editing function in the MDPI magazine to assist in the overall changes, which we hope will improve.
